# Alterations in the Anatomy and Ultrastructure of Leaf Blade in Norway Maple (*Acer platanoides* L.) Growing on Mining Sludge: Prospects of Using This Tree Species for Phytoremediation

**DOI:** 10.3390/plants13101295

**Published:** 2024-05-08

**Authors:** Magdalena Krzesłowska, Mirosław Mleczek, Aleksander Luboński, Karolina Weręża, Adam Woźny, Piotr Goliński, Sławomir Samardakiewicz

**Affiliations:** 1Department of General Botany, Faculty of Biology, Adam Mickiewicz University, Uniwersytetu Poznańskiego 6, 61-614 Poznan, Polandadam.wozny@amu.edu.pl (A.W.); 2Department of Chemistry, Poznań University of Life Sciences, Wojska Polskiego 75, 60-625 Poznan, Poland; miroslaw.mleczek@up.poznan.pl (M.M.); piotr.golinski@up.poznan.pl (P.G.); 3Laboratory of Electron and Confocal Microscopy, Faculty of Biology, Adam Mickiewicz University, Uniwersytetu Poznańskiego 6, 61-614 Poznan, Poland; aleksander.lubonski@amu.edu.pl (A.L.); slawomir.samardakiewicz@amu.edu.pl (S.S.)

**Keywords:** trace metals, toxic elements, water stress, chloroplast, cell wall, nucleus

## Abstract

Alterations in leaf architecture can be used as an indicator of the substrate toxicity level as well as the potential of a given plant species in the phytoremediation of polluted areas, e.g., mining sludge. In this work, we demonstrated, for the first time, the nature and scale of alterations in leaf architecture at the tissue and cellular levels occurring in Norway maple growing on mining sludge originating from a copper mine in Lubin (Poland). The substrate differs from other mine wastes, e.g., calamine or serpentine soils, due to an extremely high level of arsenic (As). Alterations in leaf anatomy predominantly included the following: (1) a significant increase in upper epidermis thickness; (2) a significant decrease in palisade parenchyma width; (3) more compact leaf tissue organization; (4) the occurrence of two to three cell layers in palisade parenchyma in contrast to one in the control; (5) a significantly smaller size of cells building palisade parenchyma. At the cellular level, the alterations included mainly the occurrence of local cell wall thickenings—predominantly in the upper and lower epidermis—and the symptoms of accelerated leaf senescence. Nevertheless, many chloroplasts showed almost intact chloroplast ultrastructure. Modifications in leaf anatomy could be a symptom of alterations in morphogenesis but may also be related to plant adaptation to water deficit stress. The occurrence of local cell wall thickenings can be considered as a symptom of a defence strategy involved in the enlargement of apoplast volume for toxic elements (TE) sequestration and the alleviation of oxidative stress. Importantly, the ultrastructure of leaf cells was not markedly disturbed. The results suggested that Norway maple may have good phytoremediation potential. However, the general shape of the plant, the significantly smaller size of leaves, and accelerated senescence indicated the high toxicity of the mining sludge used in this experiment. Hence, the phytoremediation of such a substrate, specifically including use of Norway maple, should be preceded by some amendments—which are highly recommended.

## 1. Introduction

Mining wastes, including post-flotation wastes and mining sludges [1], are by-products of the mineral processing of ore, usually stored in the open air, often without any remediation treatment. It is generally accepted that they have harmful effects on human health and the environment, resulting in, e.g., a sharp decline in biodiversity [1,2,3]. The negative effects on living organisms involve a high content of toxic and potentially toxic (toxic in excess) elements (TE) including trace metals, e.g., lead (Pb), cadmium (Cd), mercury (Hg), aluminium (Al), copper (Cu), zinc (Zn), chromium (Cr), and metalloids, e.g., arsenic (As) and thallium (Tl) [2,3,4]. In addition, mining wastes have high salinity and alkalinity (pH above 9), low water retention capacity, and deficiencies in soil organic matter and the main minerals necessary for plant growth, e.g., N, P, and K. Mining wastes also show physical structures that are unfavourable for plant growth [2,3,5]. The world annual production of mining wastes exceeds 10 billion tons [2,3]. Hence, the restoration of mining wastes remains a serious problem [2,3,6]. Therefore, phytoremediation, involving the use of plants for removing TE and/or reducing the TE toxicity of polluted areas, including mining wastes, is still at the centre of scientific and practical economic interests [2,5,7,8,9,10]. Phytoremediation is a relatively cheap, ecologically friendly “green technology” and can be applied to the remediation of relatively large, contaminated areas [2,7,8].

It is generally accepted that plants used for the phytoremediation of mining wastes should be characterised by fast growth, a large increase in biomass, a dense rooting system, and relatively high resistance to such a harsh environment [2,8,9,11]. Most favoured are native plant species that are well adapted to specific climatic conditions [2,5,8]. One of the most suitable groups of plants for the phytoremediation of mining wastes are trees, which show most of the traits mentioned above [6,11,12,13]. It is worth noting that reforestation is considered the most beneficial type of reclamation of mining wastes and post-mining areas because it partially mitigates some of the impacts across the landscape and increases biodiversity, soil ecosystem functioning, and biomass production over time [6,11,14]. To date, there have been a number of notable examples of the successful reforestation of areas that have been damaged by mining wastes [5,6,15].

The phytoremediation ability of a given plant species can be monitored by observing visible injuries to its foliage [11,16,17]. However, to our knowledge, there are limited data concerning the structural alterations in the leaves of plants treated with TE. Only few data demonstrating the leaf anatomy of plants, mainly herbaceous, growing on mining sludge are available, and almost no data are available concerning the tissue arrangement of leaves in trees growing on such a substrate. In the leaves of plants exposed to TE, the most common alterations found so far have been chlorotic and/or necrotic spots [13,16,17,18,19,20]. In leaf blade anatomy, commonly occurring alterations include the thickness of leaf tissues, mainly the upper and lower epidermis, as well as palisade and spongy parenchyma and changes in the number of stomata [17,20,21]. Such alterations have also been detected in herbaceous plants that naturally colonize mining wastes—known as metallicolous plants, e.g., *Dianthus carthusianorum* [22], *Silene vulgaris* [23,24,25], and *Anthyllis vulneraria* L. [26]. Moreover, in the leaves of trees such as willows, poplars [18,19], and Sycamore maple [27] exposed to a mixture of Cd, Cu, Zn, and Pb, some groups of death cells were detected in the lower epidermis [18,19,27]. Groups of death cells were also observed in palisade mesophyll, e.g., in *Populus tremula* leaves exposed to a mixture of Cu, Zn, Pb, and Cd [19] and in *Nicotiana tabacum* treated with an excess of Zn [28]. Interestingly, the death cells of palisade parenchyma in *Nicotiana tabacum* were found to be a sink for Zn and appeared as a result of the programmed cell death process. Hence, their formation was not a symptom of injury but rather part of the plant defence strategy [28,29,30,31].

At the cellular level, in plants exposed to TE [16,19,27,32] and growing on mining sludge [26], the increase in cell wall (CW) thickness was observed in different leaf tissues [16,19,26,27,32]. It was commonly associated with a rise in the level of CW components able to bind TE ions such as phenolic compounds [16,27,32] and pectins [16,19,26]. Moreover, cell vacuolization, more dense nucleus structure, and alterations in chloroplasts were commonly demonstrated mainly in palisade and spongy mesophyll cells [18,19,27,33,34,35,36,37]. The appearance of these symptoms was linked to accelerated cell senescence (ACS) in the leaves of plants exposed to TE [18,37].

The results presented in this paper are a continuation of our earlier studies concerning the phytoremediation potential of native deciduous tree species growing on mining sludge originating from a copper mine in Lubin (Poland) [1,13]. The earlier experiments included *Acer platanoides* L. (Norway maple)—the object of the present research—which is often used for urban landscaping because it is considered to be tolerant to different ecological conditions [17,38,39]. The results of our earlier research showed that Norway maple was found to be the most promising tree species for the phytoremediation of TE from polluted areas, like mining sludge [13]. Therefore, it was selected for further studies concerning mostly physiological alterations in leaves that showed, among others, the following signs: visible signs of chlorosis [13], a significant decrease in chlorophyll b content, an altered ratio of chlorophyll a:b, an increase the content of polyphenols, and, moreover, a marked reduction in leaf area [1].

Taking into consideration the very limited amount of data about leaf structural alterations in trees growing on mining wastes, as well as the results obtained in our earlier research concerning the leaves of Norway maples growing on mining sludge, the aim of this study was to determine how growth in such harsh conditions affects leaf architecture. We put forward the hypothesis that the nature and the scale of structural alterations in the leaves of Norway maples growing on mining sludge could be a useful indicator of both (1) the plant’s ability to cope with these stress conditions and the prospect of the application of this tree species for phytoremediation and (2) the level of substrate toxicity. To test the hypothesis, we carried out the following: an analysis of alterations in leaf blade tissue arrangement and an analysis of leaf cell ultrastructure, in particular, chloroplasts. To our knowledge, this is the first paper to demonstrate the structural alterations in the leaves of Norway maple trees growing on mining sludge extremely contaminated with As 18022 [mg kg^−1^ DW] [13], differing from mining wastes such as calamine (high levels of Pb, Zn, and Cd) or serpentine soils (high levels of Ni and Cr [24] used in other works, which were mainly concerned with metallicolous ecotypes of herbaceous plants [23,25,26].

## 2. Materials and Methods

### 2.1. Experimental Design

The experimental design was described in detail in our previous paper [13]. In brief, the materials for the experiments were *Acer platanoides* L. (Norway maple), selected from over 2000 specimens of two-year-old leafless seedlings obtained from the Pniewy Forest Division (52°29′4″ N; 16°15′28″ E) in March 2017 and in 2018. Norway maple plants with similarities in their development of root systems and stems with leaves as well as plant biomass were planted in cylindrical white pots (21 × 21 cm, diameter and height) in the following groups: 25 specimens in mining sludge from a copper mine in Lubin, Poland, without any modifications [1,13]; and 10 plants for the control group in soil with a concentration of elements (for soil characteristics, see [13]) that was similar to the geochemical background of Polish soil [40,41]. Both control soil and mining sludge were homogenized before the experiment using a POLYMIX PX MFC 90 D/H homogenizer (Kinematica, Inc., New York, NY, USA). The pot experiment was conducted in a greenhouse from 16 March to 15 June in 2017 and in 2018. This period of time is described as the most intensive for plant growth [42]. The greenhouse was equipped with a mechanical ventilation and watering system and automatic data loggers recording ambient parameters every hour. The mean values of temperature, relative humidity, and the concentration of CO_2_ were 22.8C, 50.5%, and 459 ppm, respectively, with a range of 10.9–38.5 C, 25.4–78.5%, and 293–687 ppm, respectively.

### 2.2. Microscopic Analysis

#### 2.2.1. Sampling

Morphological observations concerned the scale of the development of the above-ground part of the young trees, including the size and colour of leaves. The observations were conducted using a stereomicroscope (Zeiss SteREO Discovery.V8; Oberkochen, Germany). We focused on alterations in the colour and size of the leaves.

For microscopical observations, we always sampled the third leaf from the top of the stem of the trees growing both on mining sludge and in the control group. Next, small 3–5 mm long samples of leaves, together with the main vein near the top of the leaf, were cut off by a razor blade and rapidly placed in a glass bottle with a fixative solution (see below). Semi-thin sections (2.5 µm thickness) were cut using a diamond knife on an ultramicrotome (Leica Ultracut EM UC6, Vienna, Austria). Semi-thin sections were collected on basic glasses then dried (80 °C) and analysed using a confocal scanning laser microscope (LSM 510; Zeiss, Germany) in transmitted light.

#### 2.2.2. Transmission Electron Microscopy

Leaf anatomy and ultrastructure were analysed in leaves preserved via fixation with 2% glutaraldehyde and 2% paraformaldehyde in 0.05% cacodylate buffer in the first stage and with 1% osmium tetroxide—aqueous solution—in the second stage. After fixation, the samples were processed as previously described [43]. Dehydrated material was embedded in Spurr resin (Polysciences).

Ultra-thin sections (70 nm thickness) were cut on the same ultramicrotome as semi-thin sections and as described above (see Section 2.2.1). The ultra-thin sections were collected on Formvar-coated copper grids (Polysciences). The sections on grids were stained with 9% uranyl acetate and 0.5% lead citrate—both aqueous solutions. The staining for each solution took place over 20 min. Then, the sections were examined with a JEOL JEM1200 EX TEM transmission electron microscope (Tokyo, Japan) at an accelerating voltage of 80 keV.

### 2.3. Measurements

Measurements were carried out precisely on micrographs from a TEM microscope. We took micrographs of 3 leaves from each of the two groups of experiments (see Section 2.1). The measurements were made using the Fiji program. The Fiji program is a newerdistribiution of ImageJ. The version used was (Fiji Is Just) ImageJ 2.14.0/1.54f based on a Java 1.8.0_322 environment, installed on a Windows 11 Home operating system 64-bit version. Version of this software was accessed 29 June 2023 (https://doi.org/10.1038/nmeth.2019) [44]. The locations for measurements on the TEM micrographs were selected randomly.

Measurements included the following: (1) the width of entire leaf blade; (2) the width of individual tissues including the upper epidermis, palisade parenchyma, spongy parenchyma, and lower epidermis; (3) the area of palisade parenchyma cells; and (4) the thickness of cell walls (CWs) that were not in the region of cell junctions, including the peripheral CWs of the upper and lower epidermis with the cuticle but without outgrowths and the CWs of palisade and spongy parenchyma.

### 2.4. Statistics

The statistically significant differences were established using R Studio version 2023.09.0, with the R language in version 4.3.1 (Student’s *t*-test) and are marked with an asterisk (* *p* < 0.05, ** *p* < 0.01, *** *p* < 0.001).

## 3. Results

### 3.1. General Tree Morphology and Alterations in the Entire Leaf Blade Architecture

After 3 months, the Norway maples growing on mine tailings were still alive and had leaves (Figure 1A). However, the overall shape of the tree was reduced and showed few side branches, and the number of leaves was smaller in comparison to the control, where many side branches and leaves had developed (Figure 1A). Both plants growing on mining sludge and under control conditions showed simple, palmate, five-lobed leaves. However, in trees growing on mining sludge, the leaf surface was markedly smaller by about 50% [1], and the leaf blade edges were rather irregular and showed sinuses sharper than in the control (Figure 1B). Moreover, control leaves were more intensely green than those of plants growing on mining sludge (Figure 1B).

The micrographs from TEM showed that both in the control (Figure 2A) and in maple S (Figure 2B), the protoplasts of all cells building the leaf tissues were abundant in organelles. No dead or seriously malformed cells were visible in the maple S leaf.

The palisade parenchyma of maple C (Figure 2A) was formed mainly by one cell layer. Occasionally, two layers of cells occurred. In contrast, the palisade parenchyma in maple S contained more than one layer of cells in many places (Figure 2B).

The number of cell layers building spongy parenchyma in maple C (Figure 2A) was similar to maple S (Figure 2B).

The measurements made on TEM sections showed no statistically significant differences in the thickness of the maple S leaf in comparison to the control (Table 1). The mean thickness of the leaf blade in the control was 73.07 μm, and the mean thickness of the leaf blade of maple S was 77.5 μm. The differences between the maple C and maple S leaves mainly concerned the thickness of individual tissues and their organization. The most distinctive difference was noted in the upper epidermis. In maple S, the mean width of the upper epidermis was 15.13 μm, while in the control, it was 9.81 μm. Hence, the upper epidermis in the maple S leaf was 36% thicker than in the control (Table 1). Contrary to this, the mean width of the palisade parenchyma in the maple S leaf was 17.90 μm, while in the control, the mean width was 22.11 μm. Thus, in the control, it was 19% thicker. The thicknesses of spongy parenchyma and the lower epidermis were similar in both maple S and maple C (Table 1). The ratio of the palisade parenchyma to spongy parenchyma was 1:1.4 in the control, while it was higher in the maple S leaf at 1:1.8.

### 3.2. Alterations in the Architecture of Upper Epidermis

Generally, the cells building the upper epidermis showed a considerable variability of shapes. In the control leaf, the cells appeared as an oval (Figure 1C) or almost rectangular shape (Figure 3A). In the maple S leaf, the cells were smaller and were rectangular (Figure 1D and Figure 3D), oval (Figure 3E), or even triangular in shape (Figure 3E). Small triangular cells were located pressed between large cells that were dominant in this tissue (Figure 3E). In some places, the epidermis contained two layers of cells (Figure 1D, Figure 2B and Figure 3E). The central part of epidermal cells was occupied by a large vacuole—both in the control (Figure 3A) and in maple S (Figure 3D,E). The vacuoles, especially in maple S leaves, contained autophagic bodies (Figure 3D–F). The autophagic bodies were surrounded by a single membrane, which suggested microautophagy.

#### 3.2.1. Ultrastructure of Cell Wall

The CW located on the environmental side (peripheral CW) in the maple C leaf (Figure 3A–C), was markedly thicker than other CWs surrounding the cell. The mean thickness of peripheral CW together with the cuticle was 1.9 µm and was thicker than the peripheral CWs with cuticles in the maple S leaf, where the thickness was 1.75 µm (Table 2).

The peripheral CW of the upper epidermis in the maple C leaf was deposited regularly, and its edges were predominantly smooth (Figure 2A and Figure 3A–C). There were small, not very numerous outgrowths visible on the environmental side of the CW (Figure 3A,C). In the control, the peripheral CW was formed by three structurally different layers (Figure 3B,C): (1) the first one—mainly homogenic—was located at the protoplast and occupied approximately 30% of the width of the entire CW (Figure 3B,C); (2) the second—granular—was located above the homogenic layer and contained numerous electron-dense granules (Figure 3B,C) and vesicle-like structures (Figure 3B); and (3) the third—cuticle—was the most peripheral, homogenic, relatively electron-dense layer, covering the CW, easily visible in Figure 3C. In cell junctions, the structure of the peripheral CW differed from the rest of the CW (Figure 3A,B). In these regions of the CW, it was not possible to distinguish the three layers described above, because in cell junctions, the peripheral CW contained many electron-dense granules, fibrillar and vesicle-like structures within the entire area (Figure 3B).

In the maple S leaf, the cells of the upper epidermis were more elongated than in the control (Figure 3D). The peripheral CW of the upper epidermis was deposited irregularly, and its structure was disturbed (Figure 3D–H) in comparison to the control (Figure 3A–C). Moreover, the edge of CW was wavy (Figure 3D,F) due to the occurrence of many, quite extensive outgrowths. The outgrowths varied in size and shape (Figure 3E,G,H) but were generally markedly bigger than in the control, where CW outgrowths occurred rarely and were rather small (Figure 3A,C). Sometimes the outgrowths in maple S peripheral CW contained electron-dense deposits on the tips (Figure 3H). The cuticle was present in maple C (Figure 3B,C) and maple S leaf, but in the control, it was rather smooth (Figure 3C), while in maple S, it was wavy and covered with numerous outgrowths of peripheral CW (Figure 3G,H). At many sites, the three-layered structure of CW, characteristic of control CW (Figure 3C), was indistinguishable in the peripheral CWs of the maple S leaf (Figure 3F). In addition, numerous electron-dense granules, varying in sizes and shapes, occurred within the entire peripheral CW of maple S (Figure 3F). However, in other maple S leaves, there were regions of peripheral CW that showed a homogenic structure only, with rarely present granules, where only the cuticle was distinguishable as a more electron-dense layer (Figure 3E–H). In cell junctions of the upper epidermis of the maple S leaf, the peripheral CW, similarly to the control, showed a different arrangement (Figure 3D–H) than the rest of the CW. In these regions, the CW contained vesicle-like structures, electron-dense granules and fibres over the entire area (Figure 3D,F,H) or. in other cases, a homogenic, electron-dense structure (Figure 3E,G). In the maple S leaf, the peripheral CW was also relatively thick in comparison to other CWs surrounding the cell. However, as we mentioned above, the peripheral CW of the upper epidermis in the maple S leaf was thinner than peripheral CWs in the control (Table 2).

Anticlinal CWs of upper epidermis cells in maple C were regular and showed rather smooth edges (Figure 3A), while in maple S, the anticlinal edges were wavy and the CW width in many sites varied: at some sites, it was thin, and at others, it was thick and formed local, numerous CW thickenings, different in size and shape (Figure 3D,E,G). Local CW thickening was also present in periclinal CWs (Figure 3E).

### 3.3. Alterations in Architecture of Palisade Parenchyma

Palisade parenchyma in maple C leaves were mainly built from one cell layer (Figure 1C, Figure 2A and Figure 4A). The cells were not tightly packed, and many quite large intercellular spaces (ISs) were visible. The cells were elongated in shape and contained numerous elliptical-shaped chloroplasts (Figure 4A and Figure 5A,C). Contrary to this, in the maple S leaf, the palisade parenchyma was often formed by more than one cell layer (Figure 1D, Figure 2B and Figure 4B,C). There were some regions where the palisade parenchyma contained one layer of cells (Figure 1D and Figure 2B), but in many other places, it contained two layers (Figure 1D and Figure 4B), and sometimes even three layers of cells (Figure 4C). Then, CWs between cells often ran obliquely (Figure 4B). The cells forming the palisade parenchyma in the maple S leaf were relatively tightly packed. Moreover, they were significantly smaller (60%); the area was 54.86 µm^2^, while in the control, it was 152.66 µm^2^ (Table 1). Furthermore, the cells were rather more rounded in shape (Figure 4B and Figure 5B) than in the control leaf (Figure 4A and Figure 5A). This resulted in more tightly packed organelles (Figure 4B and Figure 5B) in comparison to the control (Figure 4A and Figure 5A). In the maple C leaf, the cells of the palisade parenchyma were strongly vacuolized (Figure 4A and Figure 5A). The central part of the cells was usually occupied by a large vacuole, while maple S palisade parenchyma cells contained several small vacuoles (Figure 4B and Figure 5B). In the control, the vacuoles held a large number of small, granular or fibrillar electron-dense deposits (Figure 4A and Figure 5A). Sometimes, big crystals were also present (Figure 5A). In the vacuoles of both maple C and S, some structures characteristic of autophagy were visible. However, they were more often present in the vacuoles of the control cells, and they were more diversified in structure there; for example, they were surrounded by a single membrane, characteristic of microautophagy (Figure 4A and Figure 5A), or structures formed by multiple, concentric membranes, suggesting macroautophagy (Figure 4A). In maple S, only the autophagic bodies surrounded by a single membrane were visible in vacuoles, but they occurred relatively rarely (Figure 4B).

Cell walls in palisade parenchyma were regular and had smooth edges both in maple C (Figure 4A and Figure 5A) and maple S (Figure 4B and Figure 5B). The CWs were also of similar thickness in both variants (Table 2).

#### 3.3.1. The Most Altered Organelles of Palisade Parenchyma Cells

##### Chloroplasts Ultrastructure

In the control, the chloroplasts of palisade parenchyma cells showed a regular, elliptical shape (Figure 4A and Figure 5A,C). The thylakoid system was arranged parallel to the long axis of the plastid (Figure 5C). The thylakoid membranes ran straight, regularly; thus, the grana and stroma lamellae were easy to distinguish (Figure 5C). No swollen, wavy thylakoids were detected (Figure 5C). However, some chloroplasts contained relatively large and numerous plastoglobuli (Figure 4A and Figure 5A).

The chloroplasts present in the palisade parenchyma cells of maple S (Figure 5B,E) were among the most altered organelles in comparison to the control (Figure 4A,C). They were also elliptical but more rounded than in maple C (Figure 4B and Figure 5B,D,E). The thylakoid system was predominantly arranged parallel to the long axis of the chloroplast—similarly to the maple C leaf (Figure 5C).

However, thylakoid membranes, in particular the stroma lamellae, ran irregularly, the thylakoids were swollen, and the membranes were wavy (Figure 5D). Such alterations were also observed in some grana (Figure 5B). Moreover, the chloroplasts of maple S parenchyma cells contained more numerous and larger plastoglobuli than the control (Figure 4B and Figure 5B). On the other hand, in maple S palisade parenchyma cells, besides chloroplasts which showed marked alterations in ultrastructure (Figure 5D), almost intact chloroplasts occurred. In such chloroplasts especially when grana membranes ran regularly (Figure 5E). Interestingly, chloroplasts with different degrees of alteration could be present in one cell of palisade parenchyma (Figure 5B).

##### Ultrastructure of Nucleus

In the leaves of maple C, the nuclei (N) of palisade parenchyma cells were almost spherical, appearing as circular on the TEM sections (Figure 4A and Figure 5A). The nucleoplasm contained numerous granularities. It was easy to distinguish electron-dense heterochromatin from the rest of the nucleoplasm, which showed lower electron density (Figure 4A and Figure 5A). This allowed us to classify the nucleus as a chromomeric nucleus type. The nuclei of maple S palisade parenchyma cells appeared as slightly irregular in shape on the TEM sections (Figure 4B and Figure 5B). Their nucleoplasm was very dense, containing many granules. Heterochromatin, in these nuclei, was indistinguishable (Figure 5B). However, the nuclei in both variants were surrounded by a regular envelope without any visible distortions.

### 3.4. Alterations in Architecture of Spongy Parenchyma

In the spongy parenchyma of the control leaf (Figure 1C and Figure 6A), the cells were loosely distributed, and large intercellular spaces (ISs) occurred. Contrary to this, the cells of spongy parenchyma in the maple S leaf were relatively tightly packed (Figure 1B and Figure 6B). Only small ISs were visible, and sometimes almost no ISs occurred (Figure 6B,D).

In both variants, the mean thickness of the CWs surrounding the cells of the spongy parenchyma was similar (Table 2). Large-sized vacuoles occupied the centre of the spongy parenchyma cells, but in the maple S leaf, they predominantly contained electron-dense deposits (Figure 6B). Chloroplasts in maple S were more rounded than in the control and showed similar distortions in the thylakoid system to chloroplasts in the palisade parenchyma (see Section Chloroplasts Ultrastructure). In addition to chloroplasts altered in the ultrastructure, intact chloroplasts were also present in spongy parenchyma cells (Figure 6B). Nucleus alterations in the spongy parenchyma of the maple S leaf were similar to those detected in palisade parenchyma cells (see Section Ultrastructure of Nucleus).

### 3.5. Alterations in Architecture of Lower Epidermis

In the control leaf blade, the cells forming the lower epidermis were rectangular in shape, and their longer axis was arranged parallel to the section of the leaf (Figure 2A and Figure 6C). They were relatively narrow, especially near the region of the vascular bundle (Figure 2A). In the maple S leaf, the cells of the lower epidermis were also rectangular in shape, but the longer axis of the cells was arranged perpendicularly to the leaf section (Figure 2B and Figure 6D). Similar to the upper epidermis, in some places of the lower epidermis in the maple S leaf, small triangular cells were located between big cells, building this tissue (Figure 6D).

The central part of the cells both in maple C (Figure 6C) and maple S (Figure 6D) was occupied by a large vacuole, sometimes containing electron-dense deposits.

The peripheral CWs of the control cells were deposited regularly. The CW structure contained three structurally varied regions (Figure 6C), similarly to the peripheral CW of the upper epidermis (see Section 3.2.1). Moreover, control peripheral CWs were characterized by smooth edges; no outgrowths and no local thickenings were observed. The cell wall was covered by an easily distinguishable cuticle, which showed higher electron density than the CW. It was rather smooth; no visible outgrowths occurred (Figure 6C). Conversely, in the maple S leaf, the peripheral CW was deposited irregularly and was of varied thickness (Figure 6D,E). It was possible to distinguish three layers in CW structure, like those described above in the control CW (see Section 3.2.1). However, the homogenic layer, located at the protoplast, was very thin. The second layer was wide and contained many granular, fibrillar and vesicle-like structures (Figure 6E), similar to the CW structure in cell junctions. The cuticle was distinguishable, but unlike the control, it showed lower electron density than CW (Figure 6E). The peripheral CW of the maple S leaf had many outgrowths in some regions, but in other regions, it was quite smooth (Figure 2B and Figure 6B). Measurements of the peripheral CW thickness with the cuticle in the control and maple S leaves revealed a similar thickness in both variants (Table 2). The anticlinal CWs of the lower epidermis in maple S, showed wavy edges in comparison to the rather smooth edges in the control. Moreover, local CW thickenings occurred, mainly in anticlinal CWs (Figure 6D). However, there were leaves where anticlinal CWs of the lower epidermis were structurally similar in both maple S (Figure 2B) and maple C (Figure 2A).

To sum up, we detected many alterations in the architecture of the maple S leaf in comparison to the control, both at the tissue and ultrastructural levels of organization. The main alterations are summarized in Table 3. However, it is worth underlining that generally, the maple S leaf did not manifest any severe or lethal distortions in its cell ultrastructure.

## 4. Discussion

### 4.1. Alterations in the Leaf Anatomy

The most noticeable alterations in the leaf anatomy of Norway maple growing on mining sludge in comparison to the control leaf included the following: (1) a significant increase in the width of the upper epidermis; (2) a significant decrease in spongy parenchyma thickness; (3) a higher compactness of leaf tissue organization, in particular palisade and spongy parenchyma, with very small intercellular spaces; (4) two and even three cell layers of palisade parenchyma (one in control)formed by significantly smaller cells. This resulted in a markedly smaller surface of the maple S leaf in comparison to the control. Interestingly, the thickness of the entire leaf blade in maple growing on mining sludge and in the control was similar. This differs from the responses of other plant species exposed to TE, described in other papers where alterations in leaf thickness were commonly detected [20,36,45]. They were usually related to a decrease in palisade parenchyma width, similarly to the leaf of Norway maple growing on mining sludge in our experiment and/or reduction in spongy parenchyma. Such a reaction was found in, e.g., *Brachiaria decumbens* growing on soil contaminated with Zn, Cd, and Cu [46]; *Phaseolus vulgaris* [33]; *Schinus molle* [47] and *Cicer arientinum* [48]; all plants exposed to high concentrations of Cd; *Heliantus annuus* treated with Sb [49]; *Brassica napus* [50]; and *Arabidopsis thaliana* [51] in response to As. It is worth noting that, for example, in plants exposed to Cd, a decrease in leaf blade thickness was regarded as an injury symptom characteristic of the response of Cd-sensitive plants [33,47,52]. In contrast, thicker leaves, in particular thicker palisade mesophyll, in plants exposed to TE were characteristic of resistant plants [48,52,53,54]. For example, thicker leaves, particular a thicker palisade mesophyll, were described in *A. platanoides* growing in air-polluted sites in Belgrade [17]. Moreover, *S. molle*, as mentioned above, in low concentrations of Cd (20 µM) showed an increase in palisade and spongy mesophyll thickness and simultaneously an increase in the whole leaf blade. This was related to an increase in photosynthesis efficiency and evidence that *S. molle* could successfully cope with low concentrations of Cd [47]. An increase in leaf blade thickness and the width of palisade and spongy parenchyma have been recently shown to be a response to Pb in *Peltophorum dubium* [55], a Cu-tolerant tree species used in the reforestation of degraded areas [56,57] due to its adaptive capacity in polluted soils [55]. All the data show that the alterations in leaf anatomy of different plant species in response to the same or different TEs are not uniform and reveal a wide variability. Generally, phenotypic plasticity is one of the major means by which plants can cope with environmental factor variability [58]. This depends on a range of factors like the age of the plant and leaf, TE concentration, TE speciation, time of treatment, etc. [20].

However, plants growing on mining sludge have to cope not only with the toxicity of TE present in the substrate but also with high pH, salinity, and water deficiency [2,3,5]. These factors should be taken into account when comparing the reactions of plants growing in such conditions [59]. Therefore, it is interesting to compare the results obtained for Norway maple growing on mining sludge in this work, with the responses of plants naturally colonizing mining wastes and post-mining areas. It is worth noting that the data concerning the leaf anatomy of plants growing on mining sludge are rather scarce. As we mentioned in the Introduction, most available data mainly concern herbaceous plants, naturally inhabiting such areas—metallicolous ecotypes (relatively resistant to such harsh conditions). In particular, species that can tolerate a high concentration of TE include *Vicia faba* [60], *Dianthus carthusianorum* [22], *Silene vulgaris* [23,24,25], *Anthyllis vulneraria* [26], and *Rumex acetosella* [59]. The traits of metallicolous ecotypes are often comparable with those of non-metallicolous ones. For example, the leaves of the metallicolous ecotype of the legume plant *Anthyllis vulneraria* [26] commonly colonizing calamine substrate were contaminated predominantly with Pb, Zn, and Cd [25], similarly to the Norway maple leaves in our experiments. They showed a thicker upper epidermis (by approximately 15%) and a thinner palisade parenchyma (by about 37%) built by smaller cells in comparison to the non-metallicolous ecotype [26]. However, in *A. vulneraria*, the spongy parenchyma and leaf blade generally were also thinner in the metallicolous ecotype. This differed from the leaves of Norway maple growing on mining sludge in our experiments, where both spongy parenchyma and entire leaf thickness were similar to the control. Furthermore, the *A. vulneraria* metallicolous ecotype showed a lower number of cell layers in the palisade parenchyma, with two in metallicolous ecotypes and three in non-metallicolous types [26]. In contrast, in the leaves of Norway maple growing on mining sludge, the number of cell layers building palisade mesophyll increased, and in many regions, it was formed by two to three cell layers, while in the control, it was formed generally by one cell layer only.

Contrary traits to those of leaves in Norway maple growing on mining sludge were found for *Silene vulgaris* L., a plant species also commonly colonizing calamine soils and one of the most tolerant plants to TE [23]. Metallicolous ecotypes of *S. vulgaris* growing both on calamine substrate [23,24] and on serpentine soils with high concentrations of Ni, Cr, and Co [25] showed significantly thicker leaf blades than non-metallicolous ecotypes. The thicker leaf blade was involved in the significantly larger thickness of palisade and spongy parenchyma and the larger size of cells building these tissues. It was understood to be an adaptation not only to high concentrations of TEs in the substrate but predominantly to water deficiency [23,25]. It is worth noting that significantly thicker leaves were also detected in *Betula pendula* growing on post-mining heaps strongly contaminated with Zn, Pb, and Cd—this tree species is known to be a pioneering species in post-mining habitats [61]. All the data referenced above demonstrate that metallicolous and pioneering plants naturally colonizing mining wastes also show a wide diversity of alterations in leaf anatomy, even when growing on similar substrates, like *A. vulneraria* [26] and *S. vulgaris* [23,25]. Hence, the symptoms of plant adaptation to colonizing similar habitats, such as mining wastes, can be varied in different plant species. Thus, at least some of the alterations detected in the leaf anatomy of Norway maple growing on mining sludge may be considered as symptoms of adaptation and/or defence response, e.g., to water deficiency stress—commonly occurring in plants exposed to TE [23,33,59,62]. As we mentioned above, in the leaves of Norway maple growing on mining sludge, the width of the upper epidermis significantly increased, and the cells of the palisade and spongy mesophylls were also very tightly packed. This resulted in a distinctly more compact leaf tissue organization, contrary to the control leaf where the intercellular spaces were markedly larger and more abundant in the intercellular spaces. A similar response was observed, e.g., in the leaves of *Phaseolus vulgaris* treated with Cd [33] and *Heliantus annuus* exposed to Sb [49]. Smaller intercellular spaces in spongy parenchyma were detected in *A. thaliana* exposed to As [51]. Furthermore, recent experiments have shown that, e.g., drought-stressed Saragolla cultivars of wheat, generally tolerant to water deficiency, exhibited a greater compactness of mesophyll cells than the more sensitive wheat cultivar vevo. This was related to, e.g., a minor loss of chlorophyll content and better photosynthetic efficiency, and hence a better adaptability to drought stress [63]. In plants growing on mining wastes, water deficiency stress is mainly the result of distortion in root structure and functions, strongly affected by TE [62,64]. It can be also caused by the low water retention capacity of such substrates as mining wastes [2,3,5]. In our earlier paper [65], we demonstrated the severity of the alterations in the structure of the tree roots of *Tilia cordata* (Miller) Linden, growing on the same mining sludge, extremely contaminated with As, as in the case of Norway maple in these experiments. Such strong root malformations almost certainly led to disturbances in water uptake and transport, resulting in water deficiency stress in trees growing on this substrate. In plants exposed to TE, adaptation to water deficiency may be manifested by more xeromorphic leaf anatomy, e.g., Cd-exposed peanut plants displayed a thick upper epidermis, palisade mesophyll, a high palisade-to-spongy thickness ratio, as well as abundant and small stomata [66]. *Astronium aveolen* showed an increase in the thickness of the upper and lower epidermis in response to Pb [67]. Xeromorphic traits, e.g., a decrease in leaf surface and an increase in upper epidermis thickness, were also observed in *S. vulgaris* growing on calamine wastes [23]. Hence, the higher width of the upper epidermis, a more compact palisade and spongy parenchyma, in the leaves of Norway maple growing on mining sludge could be symptoms of water deficiency adaptation. However, other traits of the upper epidermis and cuticle that are characteristic of leaves of plants resistant to water deficiency, e.g., thicker leaf blade and thicker CWs, in particular peripheral CW [23,55,63,66], were not present in Norway maple leaves growing on mining sludge. Hence, smaller intercellular spaces in both the palisade and spongy parenchyma as well as the significantly smaller size of the cells building the palisade parenchyma in Norway maple leaves growing on mining sludge could also be symptoms of disorders, e.g., lower cell expansion due to lower CW elasticity and lower turgor pressure, as described in the leaves of *Phaseolus vulgaris* L. cv. Contender exposed to Cd [33] and in other plant organs of a range of plant species exposed to TE [68,69].

Another distinctive alteration in the leaf anatomy of the Norway maple growing on mining sludge was the number of cell layers building the palisade parenchyma, unlike the one cell layer observed in the control. This clearly pointed to the toxic effects of TE on leaf morphogenesis, in particular, cell division and CW formation. It is believed that one of the key roles in cell division and cell wall formation is played by the cytoskeleton [70,71]. The toxic effects of TE on microtubules and actin filaments in plants, causing serious abnormalities in cell division and CW formation, were described in many papers [20,72,73,74]; They include, e.g., alterations in the number of cells and their size and array within plant organs [20,59,65]. These data strongly suggested that the formation of two to three layers of leaf palisade parenchyma in Norway maple growing on mining sludge could be the effect of abnormalities in leaf morphogenesis, in particular, cell division and CW formation caused by TE toxicity.

### 4.2. Alterations at the Cellular Level

A characteristic alteration at the cellular level in leaves of Norway maple growing on mining sludge was the occurrence of local CW thickenings, visible in some anticlinal and periclinal CWs of both upper and lower epidermis. In general, an increase in CW thickness is a commonly occurring alteration in the leaves of plants exposed to TE and has been found, e.g., in the spongy parenchyma of *Populus tremula* [19] and different leaf tissues of *Acer pseudoplatanus* [27], both of which were exposed to a mixture of Cu, Zn, Cd, and Pb in the soil; in *Salix viminalis* in response to Cd [18]; and in *Phaseolus vulgaris* L. var. *Belna* exposed to Cu [32]. Likewise, the leaves of plants growing on mining wastes showed thickened CW, e.g., *Brachiaria decumbens* growing on soil from a dump area of a zinc mine [46] and *A. vulneraria* growing on calamine substrate [26]. Enlarged CW thickness in the leaves of plants treated with TE, especially when associated with an increase in the level of phenolic compounds, is believed to be a plant defence response involved in the alleviation of oxidative stress. Phenolic compounds are able to scavenge free radicals commonly produced in plant cells exposed to TE [16,19,24,25,26,32,68,75,76,77]. As demonstrated in our earlier paper [1], the leaves of Norway maple growing on the same substrate as in this study displayed a significant increase in the level of phenolic compounds. Thus, we suggest that an increase in CW thickness—local CW thickening formation, demonstrated in this paper—together with an increase in the level of phenolic compounds, detected earlier [1], could be a symptom of a defence response— oxidative stress alleviation in Norway maple leaves growing on mining sludge. However, the formation of local CW thickenings as well as the increased level of phenolic compounds could also be indications of another defence strategy—TE sequestration in the apoplast. Plant CWs form a barrier which protects protoplasts from TE [16,20,68] and/or can immobilize TE removed from the protoplasts, e.g., via a secretion pathway [68,78]. It was demonstrated that in the leaves of plants exposed to TE, CWs were the main compartment for accumulating TE, e.g., Pb and Zn in *Vicia faba* [60], Cu in *Phaseolus vulgaris* [32], Zn in *Nicotiana tabacum* [28,29,30], and Cd in *Hordeum vulgare* [79], in addition to plants growing on mining wastes, e.g., *A. vulneraria* growing on calamine soils [26]. The binding and accumulation of TE in plant CWs is involved in the high cation exchange capacity of this compartment, related to the presence of COOH, OH, and SH groups, able to bind bivalent and trivalent cations. These groups occur in CW components such as pectic polysaccharides, hemicelluloses, lignin, and alcohols building lignin and proteins. An increase in these compounds has been found in different plant species exposed to TE [68,69,79,80,81]. This was also the case of the leaves of trees, mainly *Salix viminalis* and *Populus tremula* exposed to Cd and Zn, where an increase in CW thickness was associated with an increase in phenolic compounds and pectins in CWs [16,18]. A similar response has been documented in the leaves of the metallicolous ecotype of *A vulneralia* where the level of low methylesterified pectins, the pectin fraction predominantly binding bivalent and trivalent metal ions, increased [26]. This suggests that CW remodelling, which leads to the formation of thickened CWs and local CW thickenings, as in the leaves of Norway maple growing on mining sludge, could be also a sign of a plant defence response, involved in the increase in the CW’s capacity for TE sequestration. On the other hand, the accumulation of TE in the CW, in particular the binding of TE ions by low methylesterified pectins, strongly reduces CW elasticity [64,68,69], which can result in the inhibition of cell elongation and, consequently, a decrease in tissues and organ extension [64,68,81,82]. A decrease in CW elasticity and cell extension could also be the result of water deficiency and salinity stresses [79], both of which are stress factors affecting plants growing on mining sludge, in addition to TE [2,3,5]. As we mentioned above, it could also explain the smaller size of cells, the lower thickness of palisade mesophyll, and the higher compactness of leaf organization observed in the maple trees growing on mining sludge.

Another alteration commonly observed in the leaves of plants exposed to TE when growing on mining sludge was the damage to chloroplast ultrastructure, such as an increase in the number and size of plastoglobuli and the distortion of the thylakoid system [26,33]. Such alterations in chloroplasts, together with nucleus condensation and the increase in cell vacuolization, are considered as stress-accelerated cell senescence [18,83,84,85]. Accelerated cell senescence symptoms also occur as the result of water deficiency and/or high salinity stress [85]—commonly affecting plants growing on mine tailings apart from TE [2,3,5,64]. In the leaves of Norway maple growing on mining sludge, particularly in cells of the palisade mesophyll, we observed a more rounded shape of chloroplasts, swollen thylakoids, larger and more numerous plastoglobuli, as well as a more condensed nucleus structure—typical symptoms of the accelerating senescence process. The only symptom absent from Norway maple palisade mesophyll was the increase in vacuolization. However, the cells in the palisade mesophyll were significantly smaller than in the control. Hence, the space in the cells was limited and we suppose that it could explain the limited increase in vacuole size. Alterations in chloroplast ultrastructure, similar to those described in Norway maple growing on mining sludge, have also been observed in other plant species, e.g., *Salix viminalis* [16], *Hordeum vulgare* [37], and *Sesuvium portulacastrum* [34], which were all exposed to Cd; in *Inga subnuda* treated with Cu [85]; in *Polygonum perfoliatum*, an Mn- tolerant species, when exposed to extremely high concentrations of Mn [86] when growing on mine tailings; in *Vicia faba* [60]; and in the non-metallicolous ecotype of *S. vulgaris* exposed to Ni [24]. It is worth noting, however, that, e.g., metallicolous ecotypes of *S. vulgaris* treated with Ni did not show such alterations in their chloroplasts [24]. Thus, accelerated senescence is rather a symptom of injury to the plant leaf, although some authors interpret it as a symptom of defence response—programmed cell death—a hypersensitive reaction [16]. It is worth underlining that in leaves of Norway maple growing on mining sludge, besides the chloroplasts which showed the symptoms of accelerating senescence, there were many chloroplasts that were found to have an almost intact ultrastructure. This suggested that they could still fulfil their photosynthetic functions quite well.

## 5. Conclusions

Alterations in leaf architecture can be used as an indicator of both the capability of a given plant species for use in the phytoremediation of mining wastes as well as the level of the substrate toxicity. Overall, the research allowed us to determine and demonstrate, for the first time, the nature and the scale of modifications in the leaf architecture of the Norway maple growing on mining sludge extremely contaminated with As.

Detected alterations such as the disturbed general shape of the plant, the markedly smaller surface of leaves, and symptoms of an accelerating senescence process in palisade mesophyll cells indicated the extremely high toxicity of the substrate. However, the leaflets of Norway maple were able to survive in such conditions throughout the experiment (3 months), and they were still alive at the end. Moreover, some alterations that were found in Norway maple leaves suggested defence and/or adaptation responses to such a severe habitat. For example, local CW thickenings could be symptoms of a defence response related to the increase in CW capacity for TE sequestration. In the light of our earlier results that showed an increase in phenolic compounds in the leaves of Norway maple growing on mining sludge [1], the enlarging CW thickness detected in this work can also be considered a defence response symptom related to the alleviation of oxidative stress. Moreover, the increase in leaf compactness as well as the increase in the width of the upper epidermis suggested plant adaptation to water deficit. A promising trait in the prospect of using Norway maple for the phytoremediation of mining wastes was also the presence of many chloroplasts which showed an almost intact ultrastructure with few disorders in protoplast ultrastructure.

Taking all these data into consideration, we can conclude that Norway maple possesses good phytoremediation potential and could be a suitable tree species for phytoremediation and the afforestation of mining wastes. Nevertheless, the alterations that occurred in the leaves of Norway maple were serious and indicated that the mining sludge used in this experiment was extremely toxic and unfavourable for plant growth. Thus, the phytoremediation of such a substrate should be preceded by some amendments, e.g., the improvement of organic matter; liming; fertilization by N, P, and K; fertilization by biochar; etc. to improve its physical, chemical, and biological properties—these measures are presently commonly and highly recommended [2,3,5,6,87,88,89].

## Figures and Tables

**Figure 1 plants-13-01295-f001:**
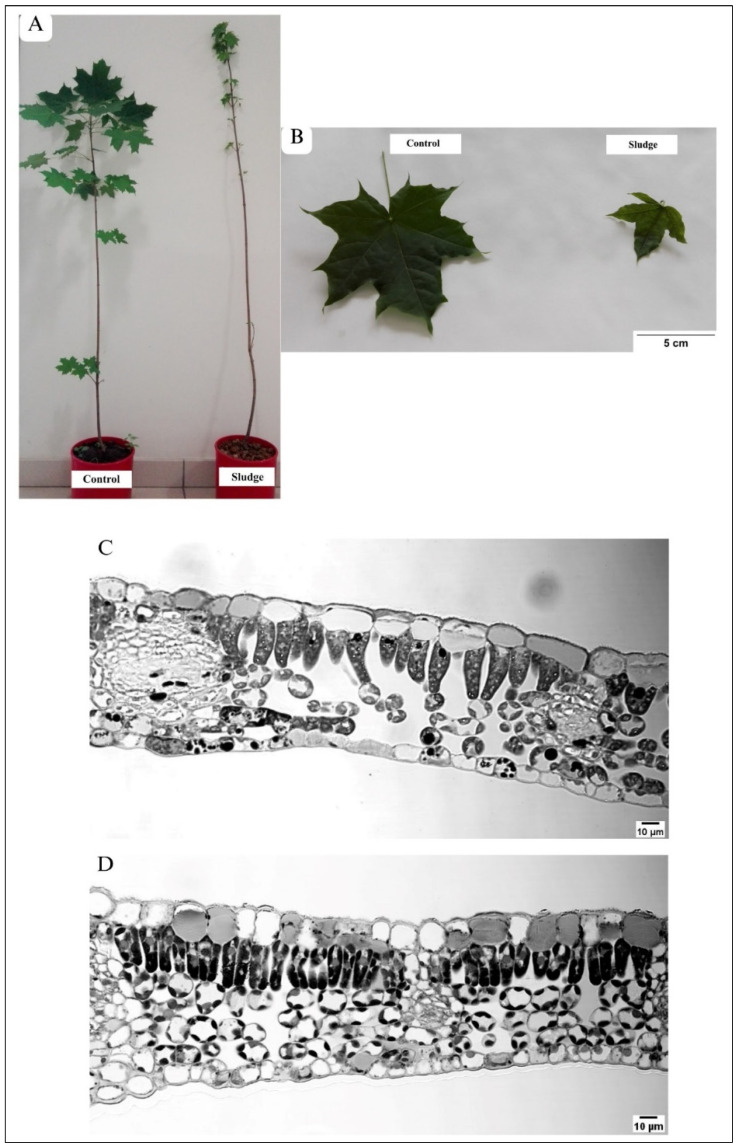
(**A**) General shape of Norway maple trees growing in control conditions and on mining sludge. (**B**) Morphology of leaf blade in maple C and maple S leaf. Light microscopy micrographs. (**C**,**D**) Anatomy of leaf bade and vascular bundles. (**C**) Maple C—loose leaf tissue organization; note the large ISs between the cells of palisade and spongy parenchyma. (**D**) Maple S—compact leaf tissue organization, note small ISs in palisade and spongy parenchyma; The microscopic analysis of leaf architecture revealed marked differences between leaves collected from Norway maples growing in control soil (maple C/control—**C**) in comparison to the leaves of trees growing in mining sludge (maple S—**D**). The differences were easily visible both in the leaf anatomy and the ultrastructure of the cells building individual tissues. The leaves of maple C (**C**) and maple S (**D**) contained all tissues that are typical in leaf anatomy: upper epidermis, palisade and spongy parenchyma, lower epidermis, and vascular bundles. No tissue was lacking or collapsed. However, the intercellular spaces, which were rather large in maple C leaf (**C**), were smaller or even absent in maple S (**D**). This resulted in a markedly more compact organization of the maple S leaf (**D**) than the control one (**C**). In the maple S leaf, the cells of the upper epidermis were visibly larger than in the control. Moreover, in many cells of the upper epidermis, the interior was filled with a grey substance (**D**) that was rarely visible in the cells of the upper epidermis in the control (**C**).

**Figure 2 plants-13-01295-f002:**
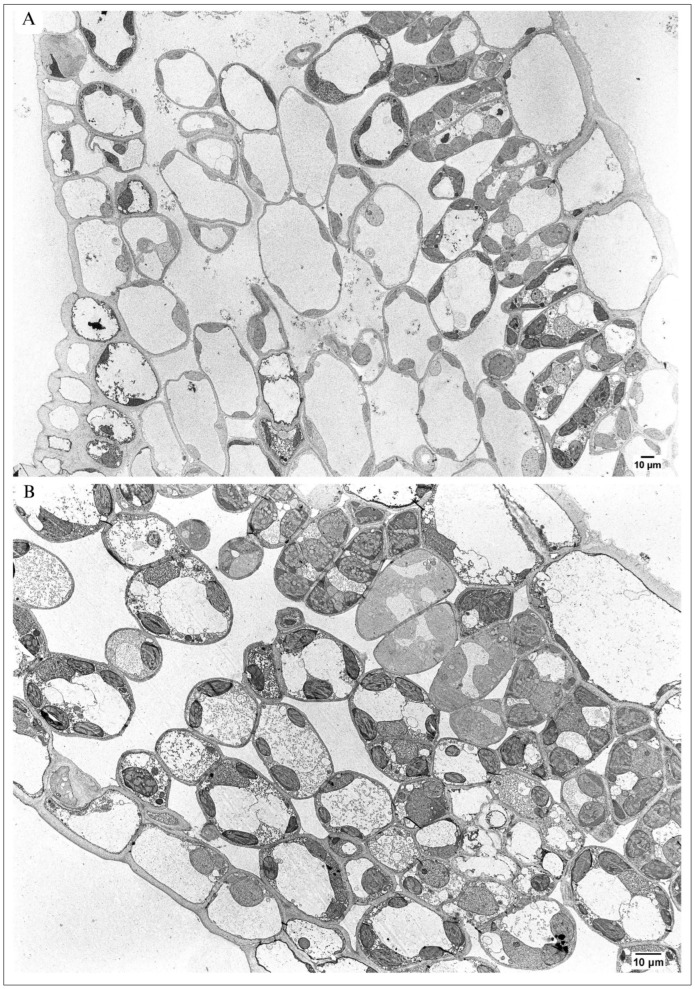
Leaf blade anatomy in TEM. (**A**) In the control leaf, note mainly one cell layer of palisade parenchyma and four to five cell layers of spongy parenchyma. (**B**) Maple S—in many regions, palisade parenchyma is built from two cell layers, and there are four to five cell layers of spongy parenchyma. Note the not strongly disturbed ultrastructure of protoplasts in all cells building the leaf.

**Figure 3 plants-13-01295-f003:**
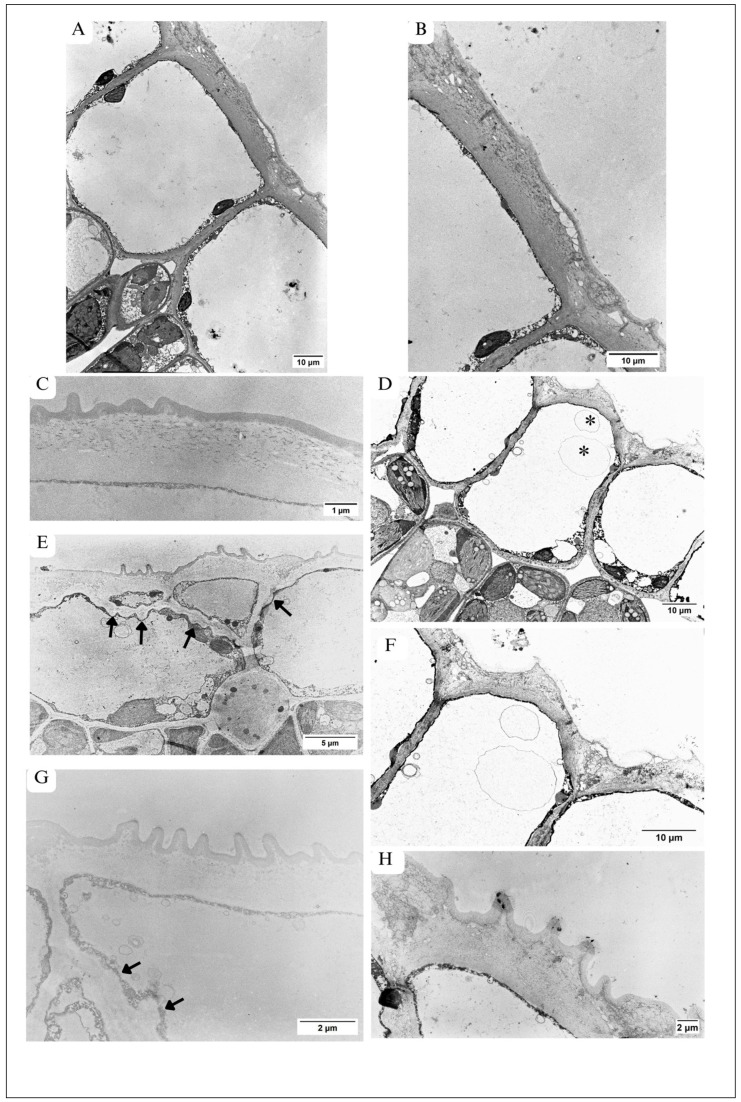
Upper epidermis of leaf—large variability in cell shapes. Ultrastructure of cell walls. (**A**–**C**) Upper epidermis in maple C. (**A**) Rectangular shape of cells with large vacuole and smooth anticlinal CWs. (**B**,**C**). Ultrastructure of peripheral CW. (**B**) Smooth edges of CW, with three different visible structural regions of the CWs: homogenic, granular, and containing vesicle-like structures. Note the different CW structure in the regions of cell junctions. (**C**) CWs with several and small outgrowths, with three easily visible and structurally different regions of the CWs: homogenic, granular, and cuticle layers. (**D**–**H**) Upper epidermis in maple S. (**D**) Rectangular cells, with large vacuole containing autophagic bodies (asterisks). (**E**) Oval and small triangular cells of upper epidermis. Note local CW thickenings (arrows) of anticlinal and periclinal CWs and autophagic bodies in vacuole (asterisk). (**F**) Magnification of (**D**); peripheral CW ultrastructure, no outgrowths, three structurally different regions present in control CW are indistinguishable here. (**G**) Peripheral CW, which shows almost homogenic ultrastructure, with many large outgrowths and easily visible cuticle. Local CW thickenings in anticlinal CWs (arrows). (**H**) Magnification of CW outgrowths containing electron-dense deposits on the tops.

**Figure 4 plants-13-01295-f004:**
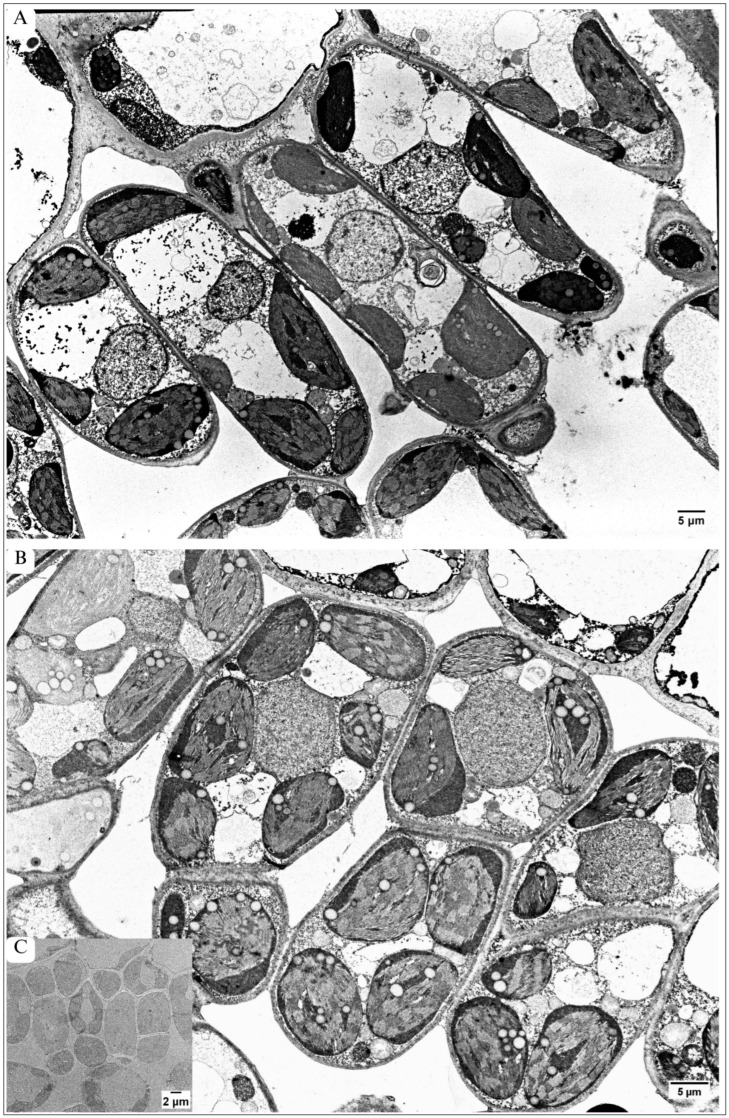
Palisade parenchyma in TEM. (**A**) In control leaf: one layer of elongated cells, relatively large ISs. (**B**,**C**) In maple S leaf: (**B**) Two cell layers of palisade parenchyma. Note small, round-shaped cells, oblique CWs between them, and small IS. (**C**) Three cell layers of palisade parenchyma.

**Figure 5 plants-13-01295-f005:**
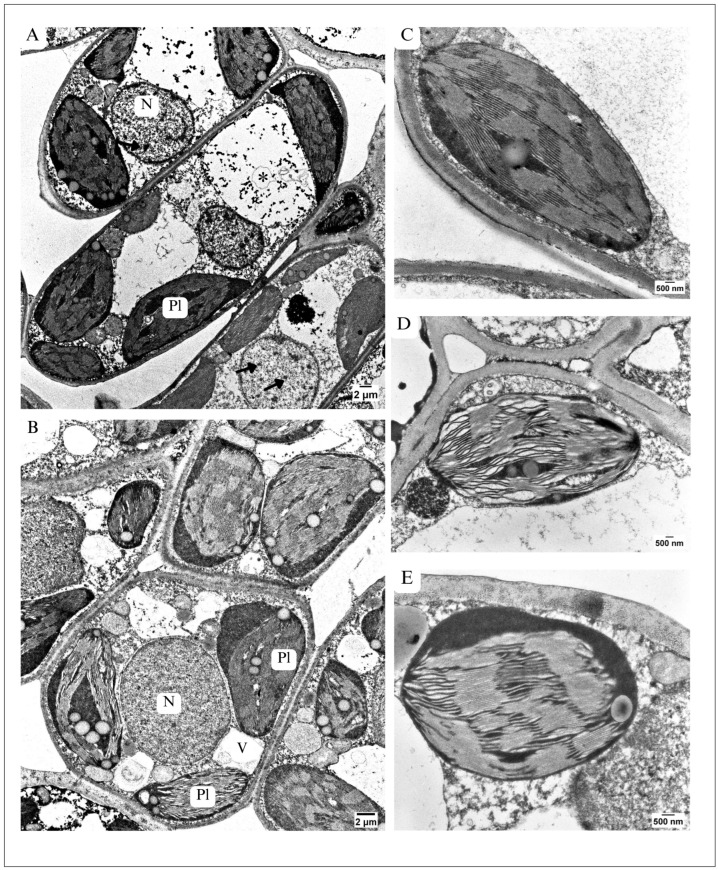
Ultrastructure of palisade parenchyma cells. Ultrastructure of chloroplasts. (**A**,**C**) In maple C: elongated cell with large vacuole(s) (V) containing autophagic bodies (asterisk) and not condensed nucleus (N) with visible chromatin (arrows). Many elliptical chloroplasts (Pl). Protoplast abundant in structures. (**C**) Chloroplast with regular thylakoid system, thylakoid almost intact ultrastructure, similar to control. (**B**) in maple S–rounded-shaped cell, protoplast abundant in structures, small vacuoles (V), dense structure of nucleus (N), chromatin indistinguishable, chloroplasts (Pl) with different level of disordered thylakoid system, wavy thylakoid membranes, big, numerous plastoglobuli, (**D**) chloroplast with serious disorders of thylakoid system, thylakoids run irregularly, commonly wavy/ swollen thylakoid membranes visible, difficult to distinguish grana from stroma. (**E**) chloroplast which shows almost intact ultrastructure, similar to control.

**Figure 6 plants-13-01295-f006:**
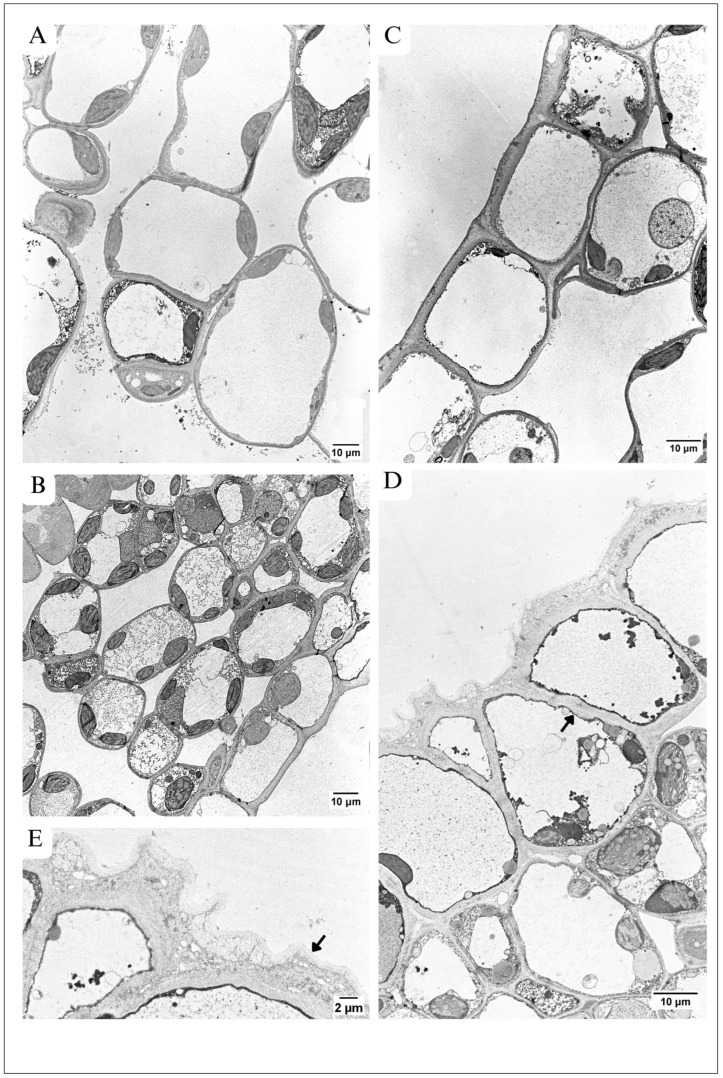
Spongy parenchyma and lower epidermis in TEM. (**A**,**B**) Spongy parenchyma. (**A**) Maple C: large IS between cells; highly vacuolized cells; elliptical, numerous chloroplasts with regular thylakoid system. (**B**) Maple S: smaller cells with large vacuoles containing electron-dense deposits. More rounded chloroplasts with large, numerous plastoglobuli and slightly disordered thylakoid system. (**C**,**E**) Lower epidermis. (**C**) In maple C: rectangular cells with smooth edge of peripheral CWs, big vacuole containing various electron-dense structures. (**D**) In maple S: different shaped cells (oval, triangular) with peripheral CW irregularly deposited and containing many vesicle-like structures. The edges of the CWs are not smooth. Local CW thickenings visible, mainly in anticlinal CWs (arrow). (**E**) Magnification of (**D**)—which shows the ultrastructure of peripheral CW of lower epidermis. Well visible irregular thickness of the CW and three layers in CW structure: the homogenic layer, located at the protoplast, the second layer–wide and contain many granular, fibrillar and vesicle-like structures–similar to the CW structure in cell junctions, the cuticle (arrow)—unlike the control, it shows lower electron density than CW.

**Table 1 plants-13-01295-t001:** Mean of the total thickness of the leaf blade, the width of the individual tissues (μm), and area of cells building palisade parenchyma (μm^2^) in Norway maple growing on mining sludge (maple S) and in control (maple C).

Measurement [µm/µm^2^]	Maple C	Significance Level for the *t*-Test	Maple S
Total thickness of leaf blade	73.07 ± 14.69	ns	73.56 ± 7.85
Width of upper epidermis	9.81 ± 2.23	***	15.13 ± 1.63
Width of palisade parenchyma	22.11 ± 3.45	***	17.90 ± 2.41
Width of spongy parenchyma	31.66 ± 6.87	ns	31.72 ± 2.9
Width of lower epidermis	9.48 ± 2.14	ns	8.82 ± 0.92
Area of cell in palisade parenchyma	152.66 ± 52.13	***	54.86 ± 20.84

Statistically significant differences between means of particular measurements in leaves of trees growing separately in control and on mining sludge, calculated by the *t*-test, are indicated by asterisks (n = number of measurements); n = 20; n = 40 for the area of parenchyma cells; mean ± SD; *** *p* < 0.001, ns—no significant differences.

**Table 2 plants-13-01295-t002:** Mean of the cell wall (CW) thickness (width on the TEM sections) in the individual leaf tissues of Norway maple growing on mining sludge (maple S) and in control (maple C).

Measurement [μm]	Maple C	Significance Level for the *t*-Test	Maple S
Peripheral CW of upper epidermis with cuticle	1.9 ± 0.32	***	1.75 ± 0.44
CW of palisade parenchyma	0.35 ± 0.13	ns	0.37 ± 0.09
CW of spongy parenchyma	0.44 ± 0.11	ns	0.41 ± 0.1
Peripheral CW of lower epidermis with cuticle	1.21 ± 0.22	ns	1.18 ± 0.25

Statistically significant differences between means of particular measurements in leaves of trees growing separately in control and on mining sludge, calculated via the *t*-test, indicated with asterisks; (n—number of measurements). n = 60; mean ± SD; *** *p* < 0.001, ns—no significant differences.

**Table 3 plants-13-01295-t003:** The main alterations in leaf architecture at the tissue and cellular levels of Norway maple growing on mining sludge (maple S) in comparison to control (maple C).

Leaf Architecture Trait	Maple C	Maple S
General plant shape	Well-developed plant: many side branches, numerous leaves, large leaf area	Disturbed plant development: only few side branches, low number of leaves, markedly lower leaf area
Leaf mesophylls organization	Loose	High compactness
Upper epidermis	Oval and rectangular shape of cells Mean width of this tissue about 9.81 µm	Oval, rectangular and triangle shape of cells Significantly larger: 15.13 µm (36%) mean width of this tissue
Palisade mesophyll	1 layer of cells, Elongated, oval-shaped cells Large IS Mean width of this tissue 22.11 µm	1–3 layers of cells; Cell size markedly (about 60%) smaller than in control Round-shaped cells Small IS Significantly lower mean width of this tissue by 19%— 17.90 µm
Spongy mesophyll	Large IS	Small IS
Lower epidermis	Rectangular shape of cells—the longer axis arranged parallel to the leaf section	Rectangular shape of cell—the longer axis arranged perpendicular to the leaf section; occurrence of triangular cells
CWs of upper epidermis	Peripheral CW - Thick: 1.9 µm - Low number of small outgrowths, - Three structurally different layers easily visible Anticlinal and periclinal CWs - Smooth edges - Regular thickness	Peripheral CW - Lower thickness by 8%: 1.75 µm - Numerous large outgrowths - Three structurally different layers indistinguishable Anticlinal and periclinal CWs - Wavy edges - Varied thickness because of occurrence of many local CW thickenings
CW of lower epidermis	Peripheral CW - Smooth edges, - Regular thickness—no outgrowths - Three structurally different layers easily visible Anticlinal and periclinal CWs - Smooth edges	Peripheral CW - Irregular edges, - Many outgrows, - Three-layered structure visible but disturbed Anticlinal and periclinal CWs - Wavy edges, varied thickness because of occurrence of many local CW thickenings
Chloroplasts	Elliptical-shaped Thylakoid membranes run straight, regularly Not very numerous plastoglobuli	More rounded shape than in control, Irregular running of thylakoid system, swollen and wavy thylakoids Numerous, large plastoglobuli
Nucleus	Nucleoplasm not highly condensed, heterochromatin easy to distinguish	Nucleoplasm condensed, heterochromatin indistinguishable

## Data Availability

Data are contained within the article.

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
