# Peer review of "Alterations in the Anatomy and Ultrastructure of Leaf Blade in Norway Maple (Acer platanoides L.) Growing on Mining Sludge: Prospects of Using This Tree Species for Phytoremediation"

_plants, 2024, doi:10.3390/plants13101295_

Round 1
Reviewer 1 Report
Comments and Suggestions for Authors
The paper “Alterations in anatomy and ultrastructure of leaf blade in Norway maple (Acer platanoides L.) growing on mining sludge in prospect of use this tree species for phytoremediation” by Magdalena KrzesÅ‚owska, MirosÅ‚aw Mleczek, Aleksander LuboÅ„ski, Weręża Karolina, Adam Woźny, Piotr GoliÅ„ski and SÅ‚awomir Samardakiewicz reports the results of exact study of leaf size, shape and structure changes caused by direct environmental stress.
The study is focused strictly on changes in leaves, using 2-years-old seedling previously activated under normal conditions and re-planted into exactly prepared mining sludge substrate. Besides the apparent size and shape differences, several interesting leaf anatomy alterations were observed and well documented by transmission electron microscopy. These results are very valuable, because plants naturally growing in such polluted conditions would have adapted from the beginning and give different results (trees from contaminated soil would be much smaller, with tiny leaves, from the start, and grow less, adequately to their conditions).
The hypothesis is viable, in good accordance with the results and nice supported by TEM photographs. Those could be a bit sharper, but it is probably the best quality authors were able to get without the use of graphical smoothing. The paper is well organised and given in good, simple scientific English. Mistakes are scarce, and just formal (i.e. line 541, two brackets, line 643, missing space). Introduction is rather comprehensive, supported by many references, and can be used as a good source of general information.
Phytoremediation is still one of hot topics and its need might increase in the future, despite current trends for safe and waste-free technologies. With global climate change, some areas of the planet won’t be as favourable for agriculture and forestry in the future, leading to increased effort for healthy environment in more suitable locations. The paper brings important information and no doubt will be interesting for readers of the Plants journal.
Mostly, it could be recommended for acceptation as it is, but let’s give the authors one more chance to smooth the manuscript as much as possible. Thus, it is recommended for acceptation after minor revision.
Comments on the Quality of English LanguageI do not have other comments to the language. It is mostly fine.
Author Response
Dear Reviewer,
Thank you very much for reviewing our manuscript. We really appreciate your positive opinion about our work and we are very glad that you found our paper interesting and valuable.
The indicated errors (lines 541 and 643) in the text were corrected as well as all similar formal errors that we found carefully reading the manuscript once again.
TEM micrographs quality was corrected using GIMP program during manuscript preparation,. They are better in Word version than in PDF.
With kind regards and best wishes – Magdalena Krzeslowska - on behalf of all authors of the manuscript.
Reviewer 2 Report
Comments and Suggestions for Authors
Thank you for giving me the opportunity to revise the MS entitled “Alterations in anatomy and ultrastructure of leaf blade in Nor-way maple (Acer platanoides L.) growing on mining sludge in prospect of use this tree species for phytoremediation” by KrzesÅ‚owska and his/her colleagues that was submitted to “Plants”. In this work the author demonstrated for the first time, the nature and scale of alterations in leaf architecture at tissue and cellular level occurring in Norway maple growing on mining sludge originated from a copper mine in Lubin (Poland). The MS submitted is suitable for Plants, and some interesting results were showed. However, there are several requirements that have to consider by the authors. In this regard, the following comments are requested to be addressed by the authors:
Comment 1: The abstract and keywords need to be modified. Trace metal and heavy metal are duplicated, just keep one.
Comment 2: The Introduction section is very messy and needs to be carefully revised.
Comment 3: The format and arrangement of the Figures and tables are very messy.
Comment 4: Please check the language of the entire text for some grammar and language errors.
Comment 5: The discussion section should be more in-depth, and some references are too outdated.
Comment 6: References should not be included in the conclusion.
Comment 7: Please carefully check the format of references, such as superscripts, capitalization, journal abbreviations, Missing page number, etc.
Comment 8: I would suggest that the authors review and include the following recent studies.
Organic-inorganic composite modifiers enhance restoration potential of Nerium oleander L. to lead-zinc tailing: application of phytoremediation[J]. Environmental Science and Pollution Research 2023, 30, 56569–56579.
Comparison of phytoremediation potential of Nerium indicum with inorganic modifier calcium carbonate and organic modifier mushroom residue to lead-zinc tailing [J]. International Journal of Environmental Research and Public Health 2022, 19(16), 10353.
Best regards,
Author Response
Dear Reviewer,
Thank you very much for reviewing our manuscript. Your comments and suggestions have helped us to improve the quality of the paper.
Reviewer’s comment 1: “The abstract and keywords need to be modified. Trace metal and heavy metal are duplicated, just keep one term.”
Author’s response. Your comment about abstract modification is very general. We made it a bit shorter. Because other reviewers did not have any comments about the abstract we did not modified it too much. In our opinion the abstract is informative for readers in this form..
The term “heavy metals” and “trace metals” include similar elements but not exactly the same. The term “trace metals” includes such toxic elements like e.g. Al - that is not heavy metal. However, we have removed “heavy metal” term and introduced “toxic elements” term. In the key words.
Reviewer’s comment 2: The Introduction section is very messy and needs to be carefully revised.
Author’s response: Thank you for your suggestion. The introduction had been revised, made shorter and more clear. It includes the following parts: (1) general characteristic of mining sludge, its influence on the environment, human health and its world production, (2) phytoremediation as the “green technology” applied for restoration of mining sludge, (3) necessary traits of plants using for phytoremediation, some advantages of using trees, (4) the most common alterations in leaf anatomy and ultrastructure found in leaves of plants exposed to TE described in earlier papers, (5) because this work is the continuation of our earlier research, we indicated some alterations, mainly in Norway maple leaf physiology, detected and described in our previous papers, (6) the last part of the introduction includes the grounds of undertaken the research presented in this paper, the aims of this work, the hypothesis and short explanation of the novelty of this topic.
Reviewer comment 3. “The format and arrangement of the Figures and tables are very messy”.
Author’s response: Thank you very much for this comment. We arranged the figures according to the description in the text of the Results. We have put the scale bars in the lower, right corner of the photo, we have standardized the font size.
Reviewer comment 4. “Please check the language of the entire text for some grammar and language errors.”
Author’s response. The manuscript had been corrected by the native speaker who always makes English correction of all our manuscripts before sending for publications. However, after the suggestion concerning language correction we sent the manuscript to another native speaker. Because, we are not native English speakers we trust native speakers who correct our manuscripts and are not able to evaluate it. If in your opinion there are still some parts of the text that still need English correction we would be grateful for indicating us these places.
Reviewer comment 5. The discussion section should be more in-depth, and some references are too outdated.
Author’s response: Thank you for this suggestion but we cannot agree with this opinion. In the discussion we wanted to avoid too many speculations which were not supported by obtaining results. However, in the discussion we included: (1) the comparison of the nature and scale of alterations detected in leaf architecture in our studies with the results obtain in other research concerning alterations in leaf architecture in plants exposed to TE and growing on mining sludge, (2) we interpreted obtained results in the aspect of harmful effects and defense strategies of plants growing on such a harsh substrate, (3) we pointed out the most possible causes of occurrences of such alterations in leaves of trees growing on mining sludge, (4) taking into consideration the nature and scale of alterations detected in leaf architecture and the fact that such alterations can be considered as the indicator of the plant ability to cope with these stress conditions as well as the level of substrate toxicity, we showed our opinion in view of the application of this tree species for phytoremediation. Because other reviewers did not suggest any modifications in the discussion we have not changed it.
In the chapter “Discussion”, but also in the “Introduction”, there are many references form the last 5 years – we especially paid attention on it. However, we also found interesting data in the papers published earlier and involved in the problems undertaken in this work. Therefore, we included them too. Moreover, some results obtained in our experiments could be discussed only with the results published in the markedly earlier papers.
Reviewer comment 6. “References should not be included in the conclusion”.
Author’s response. Because in our earlier papers we also put the references in the chapter
“Conclusions”, when we regarded it as necessary, we checked it in other papers. In many of them, published in good journals, the references were placed in chapter “Conclusions”. Therefore, we did not remove the citations from the chapter “Conclusion” in this work.
Reviewer comment 7. “Please carefully check the format of references, such as superscripts, capitalization, journal abbreviations, Missing page number, etc.“
Author’s response. Thank you for your suggestion. We have carefully checked the references once again and have introduced the corrections.
Reviewer comment 8. “I would suggest that the Author’s response review and include the following recent studies”.
Organic-inorganic composite modifiers enhance restoration potential of Nerium oleander L. to lead-zinc tailing: application of phytoremediation[J]. Environmental Science and Pollution Research 2023, 30, 56569–56579.
Comparison of phytoremediation potential of Nerium indicum with inorganic modifier calcium carbonate and organic modifier mushroom residue to lead-zinc tailing [J]. International Journal of Environmental Research and Public Health 2022, 19(16), 10353.
Author’s response. In our opinion, the results included in these papers you suggested to be included to the chapter “Discussion” are rather loosely concerned the topic of our work. They concern predominantly the increase of phytoremediation potential of Nerium oleander L. thanks to application of some mining sludge amendments. Therefore, we have included the citation of one of these papers (Environmental Science and Pollution Research 2023, 30, 56569-56579) at the end of our manuscript, where we mention about the important role of mining sludge amendments in the phytoremediation process.
With kind regards - Magdalena Krzeslowska on behalf of all authors of the manuscript
Reviewer 3 Report
Comments and Suggestions for Authors
Dear Authors,
Greetings! After reviewing your article, I have some suggestions and corrections for you.
There is a need for correct usage of terms, reconstruction of sentences, and proper usage of punctuation.
Some paragraphs are confusing and need proper arrangements. In some places, I have marked. But please check the entire manuscript for clarity.
In Table 1: What is n=20? n=40 for the area of parenchyma cells, means parenchyma cells of how many leaves? Or are they on a single leave, with 40 parenchyma cells? ;
There is NO mention of these asterisk symbols in the table, but it is in the table note *P < 0.05, **P < 0.01.
To understand, the silent features/characters of anatomical differences between the control and treated plants should be provided in table format. The description of the anatomical characters is not very significant. The results are good, but the presentation needs to be improved.
Does microscopic analysis identify any elemental deposition in the leaf cells?
A major concern is the lengthy introduction. Strict the needed information and reduce the conclusion as well. Remove references citation in conclusion.
I have mentioned the corrections in the attached file.
All the best
Regards
Reviewer

Author Response
Dear Reviewer,
Thank you very much for reviewing our manuscript and for all your comments and corrections. Most of them we have introduced into our manuscript. We appreciate them because your comments really helped us to improve the quality and intelligibility of our manuscript.
The author’s responses and explanations to the general comments
Reviewer. “There is a need for correct usage of terms, reconstruction of sentences, and proper usage of punctuation”. Some paragraphs are confusing and need proper arrangements. In some places, I have marked. But please check the entire manuscript for clarity”.
Author’s response. Thank you very much for this comment. According to your suggestion we have read carefully the manuscript and corrected the terms, punctuation, sentences and paragraphs - in such places where they could be unclear – in our opinion. We hope that now the manuscript is more intelligible for readers.
Reviewer: “In Table 1: What is n=20? n=40 for the area of parenchyma cells, means parenchyma cells of how many leaves? Or are they on a single leave, with 40 parenchyma cells? ”
Author’s response. Thank you for pointing out of this omission. We thought that it was clear that “n” meant the number of measurements. However, according to your suggestion we have added the descriptions to the tables: n = the number of measurements.
We would like to explain that we made the measurements on TEM micrographs of 3 leaves taken from two series of experiments. This information we have added to the chapter “Material and Methods”. The places for measurements were selected randomly.
Reviewer. “There is NO mention of these asterisk symbols in the table, but it is in the table note *P < 0.05, **P < 0.01.”
Author’s response. Thank you very much for this comment. The explanation is included in the chapter “Material and Methods”. However, according to your suggestion, the explanations concerning statistical significance we have added to the descriptions of the Table 1 and 2.
Reviewer. “To understand, the silent features/characters of anatomical differences between the control and treated plants should be provided in table format. The description of the anatomical characters is not very significant. The results are good, but the presentation needs to be improved.”
Author’s response. Thank you very much for this valuable suggestion. We summarized most of detected alterations in leaf anatomy and ultrastructure in the Table 3, that have been included to the chapter Results” . This undoubtedly increased the transparency of the results obtained.
Reviewer. “Does microscopic analysis identify any elemental deposition in the leaf cells?”
Author’s response. In our earlier paper, cited also in this manuscript as reference [1] -Drzewiecka, K. et al. Differences of Acer platanoides L. and Tilia cordata Mill. Response patterns/survival strategies during cultivation in extremely polluted mining sludge - A pot trial. Chemosphere 2019, 229, 589-601, we showed the concentration of TE in leaves of Norway maple growing on mining sludge and in control.
In this paper, which is the continuation of the previous works, we wanted predominantly to show alterations in leaf architecture of Norway maple growing on mining sludge in comparison to control. Therefore, for good quality of the TEM images and micrographs, the samples were fixed additionally with osmium tetroxide and the sections on the grids contrasted with uranyl acetate and lead citrate. Artificially added such elements like osmium or lead, excluded the elements level analysis in the samples - as we did it in earlier experiments, e.g. using EDS X-ray microanalysis, e.g. KrzesÅ‚owska et al., 2021 – reference [74] in this paper or in RabÄ™da I., Bilski H., Mellerowicz E.J. Napieralska A., Suski S., Wozny, KrzesÅ‚owska M., 2015. Colocalization of low-methylesterified pectins and Pb deposits in the apoplast of aspen roots exposed to lead. Environ. Pollut. 205, 315-326.
Reviewer. “A major concern is the lengthy introduction. Strict the needed information and reduce the conclusion as well. Remove references citation in conclusion.”
Author’s response. According to your suggestion we have shortened both the chapters: “Introduction” and “Conclusions”.
Because in our earlier papers we also put the references in the chapter “Conclusions”, when we regarded it as necessary, we checked this rule in other papers. In many of them, published in good journals, the references were placed in the chapter “Conclusions”. Therefore, we did not remove the citations from the chapter “Conclusion” in this work.
The author’s responses to the comments placed in the manuscript
Line 17 - we have introduced “Arsenic”.
Line 19 - the term “upper epidermis thickness” – concerns the entire epidermis layer; the epidermis cells were placed sometimes at different levels (triangle cells in leaf of Norway maple growing on mining sludge were located a bit higher, pressed between two other epidermal cells). Moreover, in some places, the epidermis was formed by two layers of cells. The term includes also the cuticle. Therefore, it is broader than just the width of the epidermal cells.
Line 23 – the term “local CW thickening” - means that the thickening had limited area. The CW was not thickened on the entire length, but only in some places - locally. We used this term also in our earlier papers, e.g. in KrzesÅ‚owska et al., Alterations of root architecture and cell wall modifications in Tilia cordata Miller (Linden) growing on mining sludge. Environ. Pollut. 2019, 248, 247-259, cited in this paper too.
Line 42 – according to your suggestion, we changed it in all possible places in the manuscript.
Line 44 and 45 - we have introduced the full names of the elements.
Line 45; we meant thallium (Tl), not titanium (Ti).
Line 48: “quotation mark” - has been removed.
Line 52 and 61: when we have made the “Introduction” shorter - these sentences have been removed.
Line 184: Thank you very much for this correction. We did not noticed that we used Polish spelling. All comma have been corrected to full stops throughout the manuscript.
Table 1. According to your suggestion in all places of the manuscript the term “external CW” was changed to “peripheral CW”.
Lines 220-231 – thank you for pointing out that this paragraph was unclear. We have corrected this paragraph and we hope that it is better understood for readers now.
Line 276 – it has been corrected to (Figs. 1C,2A,4A).
Line 610 – we have corrected it.
Lines 610-612 – it was removed from the chapter “Material and Methods”
Line 614 - we have corrected to “diameter and height”.
Line 616 – we have introduced “plants”.
Line 624 – we have introduced subscript.
643-646 – thank you for this suggestion - we have switched this paragraph to the chapter concerning preparation of the samples for observations in the light microscopy.
With kind regards - Magdalena Krzeslowska, on behalf of all authors of the manuscript